# 'You're just there, alone in your room with your thoughts': a qualitative study about the psychosocial impact of the COVID-19 pandemic among young people living in the UK

Alison R McKinlay [1], Tom May [1], Jo Dawes,[2] Daisy Fancourt [1], Alexandra Burton[1]

¹Department of Behavioural Science and Health, Department of Epidemiology and Health Care, University College London, London, UK
²UCL Collaborative Centre for Inclusion Health, Institute of Epidemiology and Health Care, University College London, London, UK

**Correspondence to**
Dr Alison R McKinlay;
a.mckinlay@ucl.ac.uk

## ABSTRACT

**Objectives** Adolescents and young adults have been greatly affected by quarantine measures during the COVID-19 pandemic, but little is understood about how restrictions have affected their well-being, mental health, and social life. We therefore aimed to learn more about how UK quarantine measures affected the social lives, mental health and well-being of adolescents and young adults.

**Design** Qualitative interview study. The data were analysed using reflexive thematic analysis, with particular attention paid to contextual factors (such as age, gender, ethnicity and health status) when analysing each individual transcript.

**Setting** Data collection took place remotely across the UK via audio or video call, between June 2020 and January 2021.

**Participants** We conducted semi-structured interviews with 37 participants (aged 13–24 years) to elicit their views.

**Results** Authors generated four themes during the qualitative analysis: (a) concerns about disruption to education, (b) missing social contact during lockdown, (c) changes to social relationships and (d) improved well-being during lockdown. Many participants said they struggled with a decline in mental health during the pandemic, lack of support and concern about socialising after the pandemic. However, some participants described experiences and changes brought on by the pandemic as helpful, including an increased awareness of mental health and feeling more at ease when talking about it, as well as stronger relationship ties with family members.

**Conclusions** Findings suggest that young people may have felt more comfortable when talking about their mental health compared with prepandemic, in part facilitated by initiatives through schools, universities and employers. However, many were worried about how the pandemic has affected their education and social connections, and support for young people should be tailored accordingly around some of these concerns.

## INTRODUCTION

When COVID-19 was formally declared a pandemic[1] many governments imposed

## Strengths and limitations of this study

► We interviewed a wide range of adolescents and young adults from across the UK about their experiences during the COVID-19 pandemic.
► Data collection took place during, and in-between several national 'lockdown' periods and we elicited their views about these different stages of the pandemic.
► We did not ask participants to provide information about sociodemographic status and it is possible the sample is biased towards those who had the digital and economic means to participate in a remote interview.

social distancing restrictions to suppress the virus, including social isolation and mobility constraints. While these measures were proposed to minimise contact and virus transmission, there is evidence to suggest that they have resulted in adverse psychological consequences across general populations.[2 3]

Lessons learnt from previous epidemics suggest that youth experience measures aimed at preventing the spread of viruses as particularly distressing[4] with the potential for some effects to be long-lasting, even after restrictions have lifted.[5 6] The effect of pandemic-related disruptions in routine may be felt more acutely by children and young people with pre-existing conditions, such as autism spectrum disorder.[7] While youth are less likely to experience severe symptomology or hospitalisation with COVID-19,[8 9] the impact of quarantine represents a significant mental health threat.[10] Emerging evidence suggests that young people have experienced the greatest decline in mental health in the first wave of COVID-19, in comparison to people in other age ranges,[11] and young

people identifying as Black and Black British have been most affected in the UK compared with other ethnic groups.[12]

Reasons for this reported decline in mental health among younger people are likely to be multifaceted. First, school closures and education disruptions remove routine, structure and opportunities for socialisation,[13] which may increase loneliness and isolation. For children or young people in need of mental health support, the closing down of schools and universities removed access to coping resources or infrastructures located in these settings (eg, mental health services and peer support). Second, the age span from adolescence to young adulthood is a particularly sensitive one as young people experience many major life transitions. Disruption to these transitions can cause uncertainty and anxiety, as documented in young adults experiencing job insecurity resultant from the pandemic[14] or students experiencing the cancellation of their studies.[13] Finally, beyond these individual factors, some adolescents and young adults faced additional stressors during COVID-19, including a decline in parental well-being, child maltreatment and bereavement, all of which may compound changes in circumstances brought on by the pandemic (eg, online learning, home schooling).[15 16] These experiences were likely felt most acutely by those already affected by socio-economic marginalisation and deprivation.[17]

In these contexts, young people and adolescents were at risk of unique psychosocial consequences from the pandemic compared with other population groups, and while others have sought to quantify the impact of the pandemic on young people's mental health, there is limited work exploring young people's individual perceptions at various stages of the pandemic. In this research, we sought to explore adolescent and young adult experiences during the COVID-19 pandemic in the UK, specifically to understand how and why social distancing restrictions affected social lives, mental health and well-being.

## METHOD

We undertook a qualitative interview study to explore the experiences of adolescents (aged 13–17) and young adults (aged 18–24) throughout the COVID-19 pandemic in the UK. This work forms part of the COVID-19 Social Study (CSS), which is a national panel survey established in March 2020, where researchers have been investigating the psychosocial well-being of people in the UK throughout the pandemic.[18]

### Participant recruitment

Eligibility criteria included: aged 13–24, living in the UK, able to speak English in order to read the study information and to understand and sign a study consent form. All participants aged 16–24 provided written informed consent. Participants aged 13–15 provided assent and a parent provided written informed consent. All participants

**Table 1** Self-reported demographic characteristics

|  | Participants, n=37 |
| --- | --- |
| **Age** | |
| 13–14 | 4 |
| 15–16 | 12 |
| 17–18 | 8 |
| 19–20 | 2 |
| 21–22 | 6 |
| 23–24 | 5 |
| **Sex** | |
| Female | 23 |
| Male | 14 |
| **Ethnicity** | |
| White British | 27 |
| Mixed Race* | 5 |
| Asian and Asian British† | 4 |
| **Qualifications** | |
| No qualifications | 9 |
| GSCE | 9 |
| A levels | 13 |
| Undergraduate | 5 |
| Postgraduate | 1 |
| **Employment** | |
| Still at school | 17 |
| At university | 14 |
| Fulltime employment | 3 |
| Apprenticeship | 1 |
| Self-employed | 1 |
| Unemployed | 1 |

*Participants who identified as Mixed Race also specified that they identify as White and Black African, White and Black Caribbean, White and North African, Other Mixed Background.
†Participants who identified as Asian or Asian British also specified that they identify as Indian, Punjabi, Pakistani and Malaysian.

completed a demographics questionnaire (see table 1 for participant demographics). No participants were previously known to the interviewers prior to recruitment. In order to make an informed decision about whether to participate, potential participants received a Study Information Sheet with further details on the study, data protection measures and links to organisations providing mental health support. All were given the opportunity to ask questions and no one who was offered an interview declined to take part. No participant response rates were recorded. All participants who participated in interviews were emailed an online £10 shopping voucher as a token of appreciation for their time.

We recruited a convenience sample informed by purposive and snowballing sampling strategies. Our purposive recruitment approach was aligned with 'quota

sampling', whereby a minimum number of participants in each group (based on age, ethnicity, living situation) were recruited to ensure we were able to learn about the richness of experience across groups.[19] In considering the evidence that certain groups have faced unique challenges during the COVID-19 pandemic,[11] we screened those who registered their interest in attempt to ensure participants were from a range of diverse backgrounds in terms of age, gender, ethnicity, education level and living situation. The study was advertised via the CSS study newsletter, through partner organisations working with young people, personal contacts and for young adults aged 18+, via social media. Although we screened participants to ensure people from multiple groups were included, the nature of our recruitment process meant our sample was ultimately recruited by convenience.[19]

## Data collection

In light of the quarantine measures in place at various points throughout data collection, all participant interviews took place remotely, either over telephone or video call. Experienced, postgraduate, male and female, qualitative health researchers conducted all interviews (AB, ARM, JD, RC, SE, TM, LB). All interviewers have previously carried out interviews about mental health and social life throughout the COVID-19 pandemic for other CSS research work.[20–22] Interviews followed a semi-structured format (range: 17–65 min, average 38), with prompts used to focus on each participant's own lived experience of the COVID-19 pandemic. For adolescents aged 13–17, interviews ranged from 17 to 46 min (average: 33 min) and for young adults aged 18–24 from 29 to 65 min (average: 44 min). As the study was designed to elicit experiences of social isolation and social restrictions on mental health, well-being and social lives, topic guide questions were informed by social integration and sense of coherence theories.[23 24] The interview topic guide included questions on participants' lives before the pandemic, understanding of and adherence to government restrictions, mental health and thoughts about the future (for question examples, see figure 1. For full list, see online supplemental file).

In order to capture some of the nuanced differences between experiences of adolescents compared with young adults, we made several key choices during data collection and analysis. We conducted data collection as two separate groups according to participant age (group 1: adolescents aged 13–17 years, group 2: young adults aged 18–24). We made this decision in order to focus on experiences that were specific to different stages of development in adolescence and 'emerging' adulthood.[25] Where possible in our results, efforts have been made to clarify

- *How do you feel about the changes that have been brought about by Covid-19?*
- *Have they had an impact on your mental health or wellbeing?*
- *What are the things most bothering you at the moment?*
- *Has the pandemic meant that you have any worries for the future?*

**Figure 1** Interview guide examples.

any similarities and differences in participant responses based on their age group. Rather than progressing in a linear fashion from stage to stage during qualitative data analysis,[26] we made a decision to begin coding before data collection had finished. This approach allowed us to reflect on the way questions were asked during interviews and use prompts to learn more about topics of relevance to the research aims.

## Data analysis

Research interviews were the only source of data collected and analysed during the study. We audio-recorded interviews, which were transcribed verbatim by a transcription service. Transcripts were double checked for anonymity and accuracy, then imported into NVivo V.12. We did not use member checking methods during the study. We conducted a reflexive thematic analysis,[27 28] and used a combination of inductive and deductive coding approaches. First, an initial coding framework was developed based on the theoretical concepts explored in the interview topic guide. This was updated with new codes that were grounded in the data identified while reading through the transcripts. ARM and TM double coded three interview transcripts in the young adult group and discussed these to ensure coder consistency. ARM and JD double coded three transcripts in the adolescent group to ensure consistency between coders in identifying topics of salience. ARM and TM then carried out line-by-line coding of interview transcripts and created the coding framework. The list of themes and subthemes were developed iteratively: ARM presented themes and subthemes during weekly internal meetings with the CSS research team, in order to gain formative feedback on the comprehensibility of codes and preliminary findings. Findings were revised and finalised based on this feedback.

## Patient and public involvement

Patients or members of the public were not involved with the design, conduct, reporting or dissemination of this research.

## RESULTS

We recruited 37 participants between June 2020 and January 2021. Participant characteristics are described in table 1. The average age of participants was 18 (range 13–24 years old). The majority were female (62%) and White British (73%). All participants lived with others (eg, housemates, family, caregiver, spouse). Most participants were in secondary school or university (84%) and living at home with their parents (78%). Sixteen participants reported that they had an existing physical and/or mental health condition.

We generated four overarching themes about the impact of the pandemic on mental health and well-being: (a) concerns about disruption to education, (b) missing social contact during lockdown, (c) changes to social

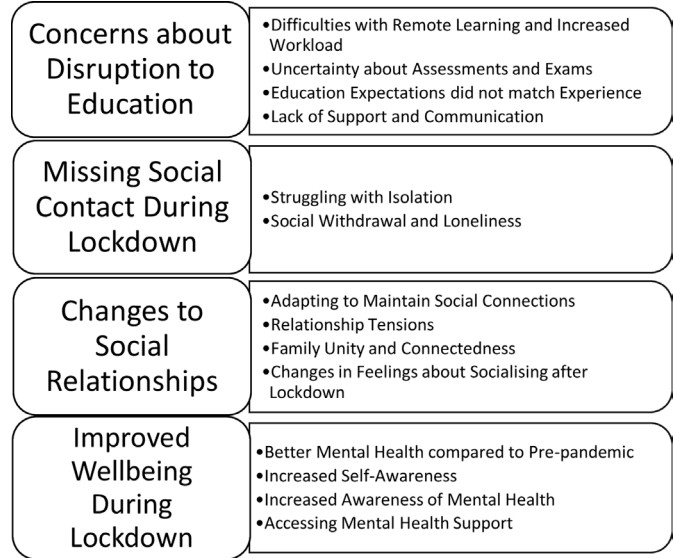

**Figure 2** Themes and subthemes.

relationships and (d) improved well-being during lockdown. Themes and subthemes are illustrated in figure 2.

### Theme 1: concerns about disruption to education
The cancellation of assessments and closure of schools and universities were described as a source of loss, concern and uncertainty for most participants. Many felt unsupported in their transition to online, remote learning and others worried about the longer-term impact of education closures on their future career and learning prospects.

#### Difficulties with remote learning and increased workload
Inability to concentrate, feelings of boredom and a lack of stimulation created difficulties with remote learning for some participants.

> I was just staring at a take home exam paper with my brain just completely blue screened. You know, just completely not functioning. I think that just general stimulus vacuum of lockdown massively contributed to that because that's an experience I've never had before. (P56, male, aged 21–22)

For those living at home with their families, having to share a workspace with multiple siblings at once, or feeling confined to a bedroom to study because everyone in the household was working from home, meant that 'days felt like night'.

> When I've sat in my room all day, there's no distinguishing between the room where I sleep and the room where I do all my work. (P20, male, aged 13–14)

After several months of social distancing and online learning, some participants felt that the move to remote online learning had increased not only their screentime but their workload as well, compared with face-to-face learning.

> My school … said we'll do live lessons, but then a teacher, instead of doing live lessons, just put the

PowerPoint online, and then we just had to take notes from that. And they'd say, make four pages of notes on this, do the questions and then that's your work done for the lesson, which wasn't very good, because instead of spending one hour a lesson, I'd spend two or three hours. (P68, male, aged 17–18)

The impact of this increased workload on future examinations and ultimately grades created an additional source of concern and uncertainty among some participants.

> I know that the virus has definitely had a big toll on how much I'm thinking about [exams], because there's so much uncertainty … My worries about exams [are] more like: 'how am I going to get the content done in time', not how am I going to be graded. (P49, female, aged 15–16)

#### Uncertainty about assessments and examinations
Many participants discussed feeling stressed, particularly at the start of lockdown, when there was a lack of clear guidance over how schools and universities would manage coursework and examinations amid education closures.

> University was probably biggest stress at the start because there was an uncertain period when we first went into lockdown, where all the universities were rushing to work out what they were going to do. At this point I was looking at handing in a dissertation and sitting three or four exams, and I didn't know if it was still happening. (P54, female, aged 20–21)

When announcements were made that examinations would be cancelled, many said that they felt upset or disappointed and some reported feeling 'robbed' of the chance to complete their education as planned.

> I'm actually really upset that I couldn't sit my exams, so yes. It was a bit of an anticlimax, because I kind of wanted all of the build-up and the apprehension, to finalise, what, two years' worth of … Well, my entire education built up to this. And when it was halted … I found it a bit difficult. (P59, female, aged 18–19)

Some participants in secondary school reported that they had studied hard outside of school hours to improve their grades even before the pandemic hit and were anxious that this additional effort would not be taken into account when considering their grade predictions.

> Not doing exams, I was gutted. I can't prove myself and I'm being given a letter that they think I could get … they didn't see how hard I was working … they don't see how hard we work outside of school … It's just annoying that we can't prove ourselves. (P34, female, aged 15–16)

#### Education expectations did not match experience
Many participants spoke of feeling disappointment that their social plans for secondary school or university had not materialised due to pandemic-related closures and

disruptions. Symbolic events tied to the start or end of education (such as 'Freshers week', 'Leavers assembly' or prom) were cancelled and described as another source of loss during the pandemic.

> We didn't get the finale of Year 13 like we normally do where there's like a prom and things like that and it was all just a bit rushed so we didn't get to say bye. (P53, male, aged 18–19)

Some said that they felt they had 'missed out' on an experience that they saw as a rite of passage and one that they had been looking forward to.

> Because we were planning on going out so much and that all being cancelled, it was awful. The day when I was meant to have prom I put my prom dress on just because I wanted to … it was like the idea of it all being cancelled I felt like I was missing out. (P55, female, aged 15–16)

### Lack of support and communication

Many participants described feeling that there was a lack of support for their mental health and well-being from education providers during the pandemic.

> The school hasn't been great in terms of providing help … the general attitude is: 'it's okay not to be okay, but we're not going to do anything about it'. I know a couple of other schools have a similar situation where you feel like, do they know that you're struggling, but they're not going to do anything. (P61, female, aged 15–16)

In cases where support was offered, some said that they had experienced this as performative rather than genuine.

> And I feel really let down by the school … Our head of year did a wellbeing check and it didn't even seem genuine, it just seemed like she had to do it for her ego or something like that … (P34, female, aged 15–16)

Others described poorly managed communication and a lack of information from education providers about pandemic-related changes, which left students uncertain about their options for rearranging assessments and coursework.

> It would have been nice if I could speak to my personal tutor or something about it, but I think everyone was so busy at the beginning trying to record lectures and put them online, and everything was a bit hectic. So, if there was support it wasn't very clear or easy to find. (P64, female, aged 20–21)

Poor communication and a perceived lack of support from education providers also resulted in some students feeling forgotten about or unfairly treated. This led to feelings of anger, frustration and a sense that their experiences were not valued.

> At the beginning, kind of March to May period, the response was absolutely shambolic and pretty malevolent to be honest … telling people to go home at their own expense up to and including sort of international students being told that they had to leave within a week even if that meant that their flights were two grand or something. (P56, male, aged 20–21)

### Theme 2: missing social contact during lockdown

Many participants reported a decline in well-being due to feeling lonely and isolated, particularly at the start of lockdown. Being unable to see friends or family and feeling that they were missing out on important life events was a source of sadness and frustration. Some withdrew themselves even further from online social activities after government restrictions were imposed.

### Struggling with isolation

Most participants said that their mental health was affected by social distancing restrictions. Many described this as being due to feeling socially restricted or 'trapped' at home and being unable to maintain their usual activities and relationships.

> I'm not a very anxious person. I've struggled with mental health anyway, but … I think in the beginning of lockdown, being trapped at home wasn't ideal in terms of my mental health. (P59, female, aged 18–19)

The feeling of being inside and at home for an indeterminate period of time was overwhelming for some, leading to a desire to overcompensate for what was perceived as 'lost time' when restrictions began to ease after the first wave of the pandemic.

> As soon as restrictions were relaxed enough that you could meet someone socially distanced, I was down at my friends' houses that are 100m down the road, three or four times a week, just to get out of my house. I got so frustrated being cooped up. (P55, female, aged 15–16)

The social distancing restrictions were especially challenging for participants who had prior mental health concerns, many of whom reported that their well-being had declined while they were socially isolating.

> … during lockdown, I got horrendously depressed. So, I've had stuff going on mentally, but I was finding it tough. The social isolation made everything crash really badly. (P39, male, aged 21–22)

### Social withdrawal and loneliness

While many participants had adapted their lives in response to social distancing restrictions, some, particularly adolescents, described socialising less during the lockdown and reported feelings of loneliness or social withdrawal.

I don't really speak to much of my mates as I used to. It's made me quite distanced with quite a lot of people. (P66, female, aged 16–17)

At the start of the lockdown, some participants felt uncertain or overwhelmed by the move to online social interactions and made a conscious decision to withdraw from this method of contact.

Before lockdown, I never used video calls or phoning people anyway. I spoke to people through messaging a lot, but I never used Zoom. I never heard of it to be honest. So, when I was suddenly forced to do that, it was just too much, so I've stopped talking to people. (P20, male, aged 13–14)

Boredom was commonly reported by a majority of participants and at times lowered their motivation to interact with others. As the pandemic went on, some participants said that they had nothing to talk about with their peers, or that friends seemed uninterested in keeping in contact. Therefore, efforts to maintain social connection ceased and usual social ties further disrupted.

I definitely tried to keep in touch with my school friends more at the start, but then it felt very one-sided, so I stopped making all the effort and I haven't spoken to a few of them in a few months. (P44, female, aged 17–18)

Participants described feeling lonely throughout different stages of the pandemic, as though the world felt smaller from being at home during the lockdown without any social contact.

There was some socialising over Zoom but overall, I think my world just sort of contracted fairly significantly … (P56, male, aged 21–22)

Several participants felt lonely even in the presence of others as restrictions began to ease (but rules such as mask-wearing and no touching remained in place), with many associating their feelings of loneliness with a lack of physical contact and closeness.

So, even though we are back at school, we're still struggling through the same things. We're still feeling lonely, that we're not able to have that physical contact, which I'm looking forward to having if the pandemic gets better. (P51, female, aged 15–16)

### Theme 3: changes to social relationships
#### Relationship tensions
Participants spoke of distress and anger regarding family and friendship tensions during the pandemic. For some, this was due to not being permitted to see their friends, which led to relationship breakdowns.

What's changed is it's put a lot of pressure on relationships, like with my boyfriend and a lot of people I know of have broken up with their boyfriends or girlfriends because of this quarantine and not being able

to see them every day, or be around them as much. (P19, female, aged 15–16)

For others, differences in opinions between friends led to arguments over what was acceptable or not when following the social distancing rules.

One of my mates, my mate, I actually ended up having an argument, because she was, like, oh, you've got to come and see me. (P66, female, aged 17–18)

Participants living with family members said that they experienced difficulties getting along with parents and siblings during lockdown, particularly when living together in small homes or when everyone was working from home.

I don't always get on with my family very well and we would be in each other's faces a hell of a lot and that got very difficult at times … I'd say it gets worse when we have to spend more time together. (P38, male, aged 18–19)

I think everyone just got a little bit stir crazy seeing the same walls and the same people. (P44, female, aged 17–18)

#### Relationship unity and connectedness
Despite relationship tensions, participants also described feeling closer with their family members throughout the pandemic and specifically during lockdown periods spent together.

My relationship with my parents is great and it's improved. It was at a great point this year and I think better than it's ever been. And I think lockdown was a good time just to spend more time with them. (P37, female, aged 23–24)

Many were reminded about the importance of family and became resolute to make more effort to connect with others in the future.

I won't take for granted as much, like just standing outside talking to my neighbour, phone calls with my [relative]. Just the little things like that. Because obviously it's so easy in normal life just to brush them aside and say, 'oh, I'll ring my [relative] later'. So now I take the time to ring my [relative] and talk to her. (P33, female, aged 1516)

Some participants had 'difficult' conversations with family during the lockdown that they otherwise might not have had, for instance, what they would do if a family member became ill with the virus. Having frank and open conversations helped some to feel more connected with their family members than before the pandemic.

I definitely think [the pandemic] forced us to talk about things that we wouldn't always want to talk about. We had to discuss as a family what would happen to my dad if he was to get ill …We discussed those things already in general because we are quite open

about that but I think it increased the intensity and frequency of those conversations about death and what would happen and health and things like that … it's helped us to be more open with each other (P41, female, aged 23–24)

### Adapting to maintain social connections

In response to social distancing measures, participants described the ways in which they had adapted their social lives during the pandemic, including trying new methods of online communication.

The idea of not knowing when you're next going to be in a crowded place, really able one on one to chat to people I think has just made people search for other ways of connecting with people and that's online. (P41, female, aged 23–24)

Some said they were using dating apps, chatting through gaming platforms and sharing COVID-related 'memes' as a way 'to cope with difficult things' and enhance their sense of social connectedness with others. Older participants in particular said that they were more focused on online dating during lockdown as a way to connect: 'I joined a dating app just to meet people'. And others sought out dating apps as a way of alleviating boredom.

I personally don't think I would've downloaded that [dating] app if it hadn't have been for the pandemic because I've tried them before and really didn't like them and hadn't had much success. And then I was just bored and felt like when am I next going to meet someone? … I feel like that probably wouldn't have happened if it wasn't for lockdown. (P77, female, aged 23–24)

New friendships were formed during lockdown with the help of chat functions on gaming and social media platforms (ie, 'WhatsApp').

I have made a bunch of friends that are from online around my age from playing a game … so during the pandemic instead of chatting with my friends I'd be playing a game with them almost every day. Well, every day for a few hours chatting, having fun which was nice. (P21, male, aged 15–16)

### Changes in feelings about socialising after lockdown

Around half of the participants described how their feelings about in-person social interactions had changed since the start of the pandemic. Some described this in terms of feeling more vigilant about being around other people and of the risks associated with future virus transmission.

I think I'm going to be a lot more cautious about other people, obviously, like people you don't know being quite close to you, physically. (P45, male, aged 15–16)

Although individual concern about their own risk of catching COVID-19 was not uniform in the group overall,

some participants reported feeling safer during lockdown away from other people and less exposed to the perceived risk of catching COVID-19.

I felt less anxious about everything when we were in full lockdown and that the lifting out of lockdown, that's quite scary in many ways for many different people … obviously introducing more risk into your life, whereas when everyone was in full lockdown, I feel like there was something quite comforting in a weird way about that. (P37, female, aged 23–24)

Adolescents in particular said they felt apprehensive about future social situations from not having been exposed to 'real life' and becoming accustomed to life in lockdown.

I just feel like I've lost the ability to form coherent sentences. I know that sounds really stupid. It's just been really hard. (P34, female, aged 15–16)

Concern about having to socialise was especially present in those returning to secondary school after months in lockdown, following difficulties in keeping in touch with friends and maintaining positive friendships.

I was nervous, because a lot changed during quarantine. I think people got really tetchy staying at home, so there were a lot of arguments. Loads of people fell out of contact with one another … So, it was harder for us to find common things to talk about, we were just speaking about the same topics again and again. So, going back [to school, I] was really nervous because we didn't know what we would talk about. (P49, female, aged 15–16)

### Theme 4: improved well-being during lockdown

For those who experienced busy and stressful lives prior to the pandemic, social distancing restrictions provided a chance to pause and reflect, with resultant mental health benefits. Having more choice about how to spend free time was described by participants as relaxing and removed previous sources of stress. For some, their well-being was enhanced by support from others and an increased public awareness of mental health self-management during lockdown.

### Improved mental health compared with prepandemic

In comparison to a majority of participants who reported struggling with their well-being during the pandemic, a smaller group of adolescents described how their mental health had improved during the first lockdown.

Well, before lockdown, my mental health wasn't great … my sleep schedule was all over the place, school stress, but actually, lockdown gave me a chance to focus and make myself a lot better in terms of the positive emotions (P49, female, aged 15–16)

They described busy or stressful school lives prior to the pandemic. School closures and a shift to online

learning, particularly for those who were not preparing for their examinations, meant that some had less work to do. Social distancing restrictions and pandemic-related closures therefore provided an opportunity to relax and reflect.

> I think it's been just I guess inconvenience rather than a full crisis in my eyes. It probably should have affected me more but, yes, I quite like it in quarantine to be honest. I don't have to get up, I can just chill around. It's a lot nicer than having to rush around constantly. (P21, male, aged 15–16)

As previously reported, the closure of schools and universities meant an increased workload for some, but the flexibility of choosing what to do between lessons and during free time meant the lockdown felt 'like a really long summer holiday' for others.

> I was happy when school got cancelled … it was still hard to do the work at home, but it was just because it was just like, in between I could just do whatever I wanted, so it was just better, sometimes I just go and play in the garden, football, or just be on my phone or the PS4. (P28, male, aged 13–14)

### Increased self-awareness

Throughout the course of the pandemic, some participants had more time to take up hobbies and exercise, but several young adults also commented on the personal growth and increased self-awareness that they had observed.

> [This experience has] taught me to take everything a day as it comes. And then see what happens. But I won't be surprised if, in a good way, that things change and it happens for the better .… it allowed me to challenge myself more … now things have started to flow better with how I am as a person. (P60, female, aged 21–22)

Some participants reflected on the stark changes in their lives during the pandemic compared with prepandemic and said they felt differently about how they wanted their life to be when social distancing restrictions eased.

> First, I think someone hit the brakes, I think, like everything stopped … [Life is] no longer that constant, hectic mess of going out … overall, it's a lot less intense maybe, now. (P39, male, 21–22)

### Increased awareness of mental health

Some participants said they felt an increased awareness of their own mental health and how to protect it during the pandemic, through initiatives driven by workplaces and education providers.

> My organisation did allocate three weeks of wellbeing leave which is a really interesting move. And that was on top of annual leave. And that was complementary leave because the reason why they did it is that they

said that it's a really hard time and they just think everyone needs the break. (P37, female, aged 23–24)

Some said they had not paid attention to their mental health in the past compared with now, having been through a pandemic and enforced social isolation.

> [the pandemic has] shown that health and mental health and mental state is important. As it always was, but I don't think I realised it as much as I would now. (P46, male, aged 15–16)

This perceived change in mental health awareness helped make it feel easier to talk to others about their well-being, and for some, this 'opened up' a dialogue with friends and family that had never been achieved before.

> It opened up really big conversations within our family about mental health that I don't think they've ever had before like the generations. So those are two really big positives for me I think. Both myself and opening that up wider … It was just conversations that we'd never had before. (P37, female, aged 23–24)

### Accessing mental health support

Some participants sought support to help cope with pandemic-related distress, either informally through friends and family, or formally through education providers, health services or employers.

> I did talk to one of my teachers actually at school, on Teams … And she helped me and a lot of friends about things to do during lockdown. (P50, female, aged 15–16)

In some cases, regular offers of support and check-ins by educational staff were sufficiently reassuring to know that help was on hand if needed.

> [University staff would] email every week, checking in. They gave me two … deadline extensions, for my work, which helped me a lot. And my tutor was really helpful. He did check in every couple of weeks to make sure I was all right. Even over the summer, when they haven't necessarily been at uni. So, that was good. (P57, female, aged 21–22)

## DISCUSSION

We sought the views of adolescents and young adults about their well-being and social lives during the COVID-19 pandemic using detailed qualitative interviews. Young people described mental health consequences associated with education closures and lockdown measures, including loneliness, frustration, a sense of loss about cancelled plans and assessments, and uncertainty about socialising in the future. Our findings highlight how some felt paradoxically isolated but also hesitant about reintegrating with others as the lockdown restrictions eased. For some, the pandemic and associated restrictions provided a platform for new relationships to develop and

existing relationships to strengthen, while for others, social isolation added strain and divisions. Despite the challenges, we found evidence of a potentially positive change during the pandemic, including an increased awareness of mental health self-management, and having more open conversations about mental health compared with prepandemic times.

While the long-term effect of school closures on young people's health, income and productivity is yet to be fully realised,[29] disruptions to education on this scale have not occurred since the second World War.[30] These changes will likely have long-term consequences requiring ongoing support,[31] with particular attention needed on how sociodemographic factors have affected young peoples' experiences during the pandemic.[32] Around 60% of students living in affluent areas had access to online learning during the pandemic, compared with 23% of students living in more deprived areas.[33] A loss of learning has already been observed by teaching staff in the UK,[34] and our findings suggest this may be linked to increased workloads, sharing small workspaces at home, and difficulties with concentration. If unaddressed, this may restrict opportunities for employment and social mobility available to young people in the future.[31] It is essential that those without the resources to work optimally from home are supported, to ensure that educational and social inequalities are not exacerbated, both as COVID-19 persists and in other future scenarios where remote learning may be required.

It is estimated more than half of students have experienced adverse psychological well-being during the COVID-19 pandemic.[35] We found that paradoxically, while some participants felt isolated and lonely, they were also apprehensive about social distancing restrictions easing and life returning to 'normal'. Graded or phased approaches for those returning to education[36] and employment are therefore recommended as lockdown restrictions ease.[37] Academic challenges have also been frequently reported by young people during the pandemic, particularly among secondary school-aged youth.[32] We noted concern among those who had access to online learning but were unable to sit examinations and finish their coursework. Several young people felt they did not demonstrate their full potential when their examinations were cancelled and did not want to be known as the generation who were given a 'free pass'. Scott *et al* also found young people reported feeling less confident and like an 'imposter' following cancelled examinations due to the pandemic.[38] Teaching staff may be well-placed to address these concerns by clearly communicating revised assessment plans and providing reassurance about the assessment process for future students affected by pandemic-related education disruptions. Teachers also require clear guidance and coordination from government and educational authorities to be able to deliver these messages,[34] which were reported to be absent during the first wave of educational establishment closures.

Young people have reported some of the highest rates of loneliness during the pandemic,[39] which have been associated with a decline in psychological well-being.[40] Initiatives that enable young people to easily connect with others and access support for their mental health while adhering to social distancing guidelines are greatly needed.[40] Participants in our study frequently cited mental health consequences due to social isolation and loneliness, which some used gaming platforms, social media and online dating apps to help address. These findings build on those shown in previous research involving young people during the first national lockdown,[41] including the sense of loss and uncertainty frequently reported by this group. Our results build on this by suggesting how social isolation led to these effects, including what felt like missed opportunities to create memories with friends and missed social events seen as rites of passage in young adulthood.

The effects of isolation and loneliness for young people are likely to persist for some time, even after social distancing restrictions have lifted.[42] The path ahead will require additional resources for education providers and coordinated efforts from policy makers, education authorities, and health and social care to provide reassurance that service provision is responsive to the needs of young people.[43] This response should be formulated in collaboration with young people to ensure that any future support incorporates appropriate coping strategies and protective activities,[44] such as those identified by our study, including check-ins from education providers and well-being initiatives from employers. These responses could help to validate the unique struggles that many young people have faced throughout the different stages of the COVID-19 pandemic.[44]

Despite the many negative effects of the pandemic on mental health in young people identified to date, evidence also suggests that young people's altruistic motivations[45 46] and prosocial behaviours have increased, including providing practical support to others and donating to charities.[47] It is encouraging that some participants in this study reported feeling more able to talk about their mental health compared with prepandemic, suggesting less social stigma perceived by young people in this study to be associated with mental health and well-being. This was, at times, facilitated by top–down efforts from education providers, employers and parents. In this study and others,[20] we found that many have been reminded of the importance of family and friends during experiences of adversity. These changes represent a potentially positive side effect of the pandemic, providing they are supported and maintained.

## Strengths and limitations

To our knowledge, this is the first UK study to use qualitative interviewing methods to learn more about the mental health and well-being of young people (aged 13–24) during multiple phases of the COVID-19 pandemic. Purposive recruitment strategies were used to try and

increase the variability of experiences within the group of participants interviewed, but despite this, our sample contained more 15 years old and 16 years old than other age groups, and participants who identified as White than other ethnic groups. Those whom we spoke with were from a range of living situations and health statuses; however, there were some data we did not collect that is likely to have influenced participant experiences during the pandemic, including socioeconomic status, sexual orientation and gender expression. Data were gathered throughout several key time points of the UK pandemic response, including as schools were reopening after the first lockdown, and as the second wave of the pandemic had begun. We explored a range of issues during interviews that were pertinent to the research questions, but some topics identified during this study could be investigated in further depth by future researchers, including the impact of the COVID-19 pandemic on intimate relationships among adolescents and young adults. We relied on participants self-identifying to take part in the research, which may have meant that the group had experiences that were different to those less comfortable discussing their experiences or with less time to take part in research. Remote interviewing techniques may have excluded some groups with limited online access; however, we also offered to conduct telephone calls and remote data collection methods enabled us to speak to more people from across the UK.

## CONCLUSIONS

When the pandemic began and restrictions were introduced, many young people lost a sense of certainty, community and belonging provided by their routines, social circles and education providers. Our findings highlight how support for young people is needed even after the virus is controlled, to help develop and maintain their social connections and access mental health support. Findings suggest that young people in the study may have felt less stigma when talking about their mental health now compared with prepandemic, which was in part facilitated by initiatives through schools, universities and employers. Most participants in our study struggled with their well-being and loneliness during the pandemic, and our results highlight the vital role of education providers giving regular guidance and support to help alleviate their concerns. Many young people we spoke with are worried about how the pandemic has affected their education and social connections, and support for young people should be tailored accordingly around these concerns. We also highlight the detrimental impact that an absence of information and support had on young people's mental health. More must be done to ensure offers of support to young people are genuine and reinforced with signposting to information, advice and services when young people do reach out for help.

**Correction notice** This article has been corrected since it was first published. The conclusion section has been corrected.

**Acknowledgements** The researchers are grateful for the support of the British Youth Music Theatre during recruitment. Many thanks to Louise Baxter, Rana Conway and Sara Esser for their help with conducting interviews. Thank you to Katey Warren and Henry Aughterson who provided feedback on the themes and subthemes. Thank you to the peer reviewers who provided critical feedback and suggestions to improve the manuscript.

**Contributors** DF and AB designed the study. AB, ARM, JD and TM collected data for the study, analysed and interpreted the data. ARM wrote the first draft with input from AB, JD and TM. All authors provided critical revisions, read and approved the submitted manuscript for publication. All authors had full access to the data in the study. All authors (ARM, TM, JD, DF and AB) take full responsibility for the integrity of this research and accuracy of the data.

**Funding** The COVID-19 Social Study was funded by the Nuffield Foundation [WEL/FR-000022583], but the views expressed are those of the authors and not necessarily the Foundation. The study was also supported by the MARCH Mental Health Network funded by the Cross-Disciplinary Mental Health Network Plus initiative supported by UK Research and Innovation [ES/S002588/1] and by the Wellcome Trust [221400/Z/20/Z]. DF was funded by the Wellcome Trust [205407/Z/16/Z].

**Competing interests** None declared.

**Patient consent for publication** Consent obtained directly from patient(s).

**Ethics approval** This research was granted ethical approval by the University College London Ethics Committee (London, Project ID: 14895/005). All participants aged 16–24 provided written informed consent to participate. Participants aged 13–15 provided written assent to participant and a parent also provided their verbal and written informed consent.

**Provenance and peer review** Not commissioned; externally peer reviewed.

**Data availability statement** No data are available. The data are not publicly available due to their containing information that could compromise the privacy of research participants.

**ORCID iDs**
Alison R McKinlay http://orcid.org/0000-0002-3271-3502
Tom May http://orcid.org/0000-0003-3077-523X
Daisy Fancourt http://orcid.org/0000-0002-6952-334X

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
