## [Reviewer comments · BMJ Open]

ARTICLE DETAILS

TITLE (PROVISIONAL)	'You're just there, alone in your room with your thoughts:' A qualitative study about the psychosocial impact of lockdown among young people during the COVID-19 panemic
AUTHORS	McKinlay, Alison ; May, Tom; Dawes, Jo; Fancourt, Daisy; Burton, Alexandra

VERSION 1 – REVIEW

REVIEWER	Scott, Stephanie Newcastle University
REVIEW RETURNED	19-Jul-2021

GENERAL COMMENTS	This is an interesting and well written paper. I do have some comments and reflections that I believe the authors should take into account that could help strengthen the manuscript. Cumulatively, these are each minor amendments, but I do feel they need particular care and attention.  The authors state on line 33, page 4, that there is limited qualitative work exploring young people's perceptions. I think some key papers are missing that should be explored in this introduction or in the discussion of the manuscript. For example:  • Scott et al (2021) – this is a different Scott et al to the one already referenced, who used qualitative interviewing in combination with diary-based methods (https://www.mdpi.com/1660-4601/18/7/3837). • Dewa et al (2021) – findings from the CCopeY study (https://www.sciencedirect.com/science/article/pii/S1054139X21000203?casa_token=rwNiTqB8zcEAAAAA:WbpEzciqqrTjXF54TwCbaM2z_nlyEnZ00y27RJUQip31yLWnKrytikkoPd eoO-KYyjOWdJENw) • Fisher et al (2021) - https://www.sciencedirect.com/science/article/pii/S2666535221000872 • O'Sullivan et al (2020): https://www.mdpi.com/1660-4601/18/3/1062 There is a big difference between the experiences of 13-year-olds and 24-year-olds. I can see this is taken into account in data collection, how is this born out within the analysis? Were there age-related differences? For example, most participants were aged 15-16? Is it worth teasing out the phenomena which is thought to be causing impact – is it lockdown, the pandemic or social distancing? It might help to tighten up phrasing e.g. restrictions imposed as a consequence of the pandemic such as social distancing etc? Phenomenology often warrants a specific approach to the data – could more detail be proved as to how this informed analysis of interview data? Strengths / Limitations –I think the sentence that this is the first UK study to use qualitative interviewing may need to be softened (see lines 23-24, page 21) – see point 1. Maybe this could be tweaked to state 'whilst several studies have used qualitative
---

	interviews etc etc, this is the first to consider the views of young people and young adults? And really emphasis the expansive age range perhaps? 6. Finally, I think the abstract could be strengthened slightly. I'd expect to see a sentence or two focusing on approach to analysis and some of the conclusion feels like new findings?
--	--

REVIEWER	Ferraz, Dulce Fundação Oswaldo Cruz
REVIEW RETURNED	21-Jul-2021

GENERAL COMMENTS	Thank you for the opportunity to read your work and congratulation on this well-crafted manuscript on an important topic. There is a lack of qualitative studies on this topic and your paper contributes to filling this knowledge gap. I recommend your paper for publication just after some minor corrections and clarifications. Below, I detail my suggestions. Abstract: I suggest following the classical subheadings – introduction, methods, results and discussion Introduction: is well-written and contextualizes the study problem. Methods: Page 5, line 9: - why is there an “i.e.” before the reference? (i.e., Bu, Steptoe, Mak, et al., 2020) Please clarify how and why you used a combination of convenience and purposive sampling strategies. One would think that you aimed to have a balanced sample in terms of participants’ demographics, but your table shows a concentration of white participants age 15 and 16yo. Also, did you consider sexual orientation and gender identity? If not, please discuss is as a limitation. You mention that “Those who returned a signed informed consent form also completed a demographics questionnaire” – not everyone returned a signed informed consent? If so, how did they consent to participate? “All interested individuals were invited to contact the study team for further information” – how many people contacted you? How many did you select? How many actually agreed to be interviewed? Did you interview all who agreed? “Potential participants received a Study Information Sheet” – why and what was the content? “Research interviews were the only source of data collected during the study” – this is a strange sentence. Why not saying “we used semi-structured interviews for data collection”? “We conducted data collection as two separate groups, to ensure any newly generated themes discussed by participants in each age group (as identified by the researcher) were sufficiently explored during data collection and analysis” – which were the two different groups? And how separating the groups helped to ensure that new themes were sufficiently explored? Results: Table 1: Do you have an explanation for why participants concentrated between 15 and 16yo? In the quotes, why not informing participants exact age? It is quite different to be confined when you are 18 or 22.
--

	Also, considering loneliness was a major struggle, did participants report how this affected their sex life and relationship with sexual partners? You briefly mention it in your results, but I was wondering if your data could allow you to develop it a bit further. Generally speaking, your data show that the pandemic had paradoxical effects on mental health and well-being – while for some it led to feeling depressed and lonely, for other it increased mental health awareness and connexion to others. I wonder if you explored if there were any demographics associated with these different perceptions and experiences. There may not be, but since you present the demographics of the group, I think it would be worth exploring it. Discussion and conclusion: Both sections are clearly presented and reflecting the study results. I suggest that you consider the limitations of not having included or identified certain groups (e.g., gay and transgender youth, those in poorer areas where schools did not manage to put online teaching in place, or who did not have the means to access this type of teaching) among which the literature shows that lockdown can have a more negative impact. Also, maybe mentioning that other potentially important topics were not explored (e.g. sexual life), if that was the case.
--	---

VERSION 1 – AUTHOR RESPONSE

Reviewer Comments: Reviewer 1

Section/Comment	Author Response
Overall	
This is an interesting and well written paper. I do have some comments and reflections that I believe the authors should take into account that could help strengthen the manuscript. Cumulatively, these are each minor amendments, but I do feel they need particular care and attention.	Many thanks to the reviewer for the valuable comments on our manuscript. We have addressed these minor revisions and summarise these below.
Abstract	
I think the abstract could be strengthened slightly. I'd expect to see a sentence or two focusing on approach to analysis and some of the conclusion feels like new findings?	Thank you for pointing out the findings in our conclusion in the Abstract, which weren't specifically outlined in the original draft. One of our key findings is that some people found it easier to talk about their mental health now compared with pre-pandemic times, which is highlighted by the conclusion sections in the Abstract: “Findings suggest that young people may have felt more comfortable when talking about their mental health compared to pre-pandemic, in part facilitated by initiatives through schools, universities, and employers.”

	We have now added the following statement regarding our analysis to the Abstract: “The data were analysed using reflexive thematic analysis, with particular attention paid to contextual factors (such as age, gender, ethnicity and health status) when analysing each individual transcript.” We have also highlighted the key findings with greater depth in the Discussion and Conclusion section (page 19 and 20). For example, this was edited in the Discussion section: “It is also encouraging that some participants in this study reported feeling more able to talk about their mental health compared with pre-pandemic, suggesting less social stigma perceived by young people in this study to be associated with mental health and wellbeing.”
Introduction	
1. The authors state on line 33, page 4, that there is limited qualitative work exploring young people’s perceptions. I think some key papers are missing that should be explored in this introduction or in the discussion of the manuscript.	Thank you very much for this feedback and for drawing these articles to our attention. The articles have been reviewed and cited where appropriate in the Introduction section.
Methods	
4. Phenomenology often warrants a specific approach to the data – could more detail be proved as to how this informed analysis of interview data?	Reference to phenomenology was intended to refer to the emphasis on learning more about each individual’s subjective experience, and questioning style during interviews (empathic, open/non-judgemental approach and use of prompts/wording to elicit individual experience). Upon reflection, the use of the phrasing here may have been overstated, so we have removed this reference and instead specified this focus under the section on data collection. “Interviews followed a semi-structured format (range: 17-65 minutes, average 38), with prompts used during interviews to focus on each participant’s own lived experience of the COVID-19 pandemic.”
Results	
2. There is a big difference between the experiences of 13-year-olds and 24-year-olds. I can see this is taken into account in data collection, how is this born out within the analysis? Were there age-related differences? For example, most participants were aged 15-16?	Thank you for drawing this to our attention – We conducted the interviews separately, so as to draw our focus on issues particular to secondary school aged people compared with working/university aged people. We also conducted the early stages of coding separately and early on (i.e., double coding), so that the coding framework was specific to each group,

	rather than considering the analysis of transcripts as one large group. Some additional text has been added on this in the Methods section: Data collection “In order to capture some of the nuanced differences between experiences of adolescents compared with young adults, we made several key choices during data collection and analysis. We conducted data collection as two separate groups according to participant age (group 1: adolescents aged 13-17 years, group 2: young adults aged 18-24). We made this decision in order to focus on experiences that were specific to different stages of development in adolescence and “emerging” adulthood (Arnett, 2000). Where possible in our results, efforts have been made to clarify any similarities and differences in participant responses based on their age group. Rather than progressing in a linear fashion from stage to stage during qualitative data analysis, (Elliott, 2018) we made a decision to begin coding before data collection had finished. This approach allowed us to reflect on the way questions were asked during interviews and use prompts to learn more about topics of relevance to the research aims.” Data Analysis “AM and TM double coded three interview transcripts in the young adult group and discussed these to ensure coder consistency. AM and JD double coded three transcripts in the adolescent group to ensure consistency across coders in identifying topics of salience.” We have also now added more text acknowledging the concentration of 15- and 16-year-olds in the sample, which may have skewed our findings. “Purposive recruitment strategies were used to try and increase the variability of experiences within the group of participants interviewed, but despite this, our sample contained more 15- and 16-year-olds than other age groups, and participants who identified as White than other ethnic groups.”
3. Is it worth teasing out the phenomena which is thought to be causing impact – is it lockdown, the pandemic or social distancing? It might help to tighten up phrasing e.g. restrictions imposed as a	Many thanks for drawing this to our attention. We have now re-reviewed the results and modified the phrasing to signal which aspects of the pandemic were said to have brought about change for the participants.

consequence of the pandemic such as social distancing etc?	
Discussion	
5. Strengths / Limitations –I think the sentence that this is the first UK study to use qualitative interviewing may need to be softened (see lines 23-24, page 21) – see point 1. Maybe this could be tweaked to state ‘whilst several studies have used qualitative interviews etc etc, this is the first to consider the views of young people and young adults? And really emphasis the expansive age range perhaps?	Thank you for this suggestion. We have now amended this sentence: “To our knowledge, this is the first UK study to use qualitative interviewing methods to learn more about the mental health and wellbeing of young people (aged 13-24) during multiple phases of the COVID-19 pandemic.”

Reviewer 2

Section/Comment	Author Response
Overall	
Thank you for the opportunity to read your work and congratulation on this well-crafted manuscript on an important topic. There is a lack of qualitative studies on this topic and your paper contributes to filling this knowledge gap. I recommend your paper for publication just after some minor corrections and clarifications. Below, I detail my suggestions.	Many thanks to the reviewer for their helpful suggestions on our manuscript. We have addressed these minor revisions in the attached file and summarise these below.
Abstract	
I suggest following the classical subheadings – introduction, methods, results and discussion	We have amended these subheadings now to Introduction, method, results and conclusions.
Introduction	
Introduction: is well-written and contextualizes the study problem.	Thank you very much for your feedback on our Introduction section.
Methods	
Page 5, line 9: - why is there an “i.e.” before the reference? (i.e., Bu, Steptoe, Mak, et al., 2020)	We initially used “i.e.,” because there are multiple publications available reporting the Covid Social Study; however, for clarity and to avoid confusion, we have now removed this.
Please clarify how and why you used a combination of convenience and purposive sampling strategies. One would think that you aimed to	We attempted to use purposive sampling strategies in order to improve the richness of our research interviews and resulting data. However, we were only able to recruit those who made contact with us and consequently, this is

have a balanced sample in terms of participants' demographics, but your table shows a concentration of white participants age 15 and 16yo.	likely to have affected our sample (i.e., those who may have been struggling more may not have contacted us, those without a computer or cellphone would have not heard about the research to register their interest). We have now rearranged the paragraph on recruitment to reflect this. “We recruited a convenience sample informed by purposive and snowballing sampling strategies. Our purposive recruitment approach was aligned with “quota sampling,” whereby a minimum number of participants in each group (based on age, ethnicity, living situation) were recruited to ensure we were able to learn about the richness of experience across groups (Campbell et al., 2020). In considering the evidence that certain groups have faced unique challenges during the COVID-19 pandemic (Fancourt et al. 2020), we screened those who registered their interest in attempt to ensure participants were from a range of diverse backgrounds in terms of age, gender, ethnicity, education level, and living situation. The study was advertised via the CSS study newsletter, through partner organisations working with young people, personal contacts and for young adults aged 18+, via social media. Although we screened participants to ensure people from multiple groups were included, the nature of our recruitment process meant our sample was ultimately recruited by convenience (Campbell et al., 2020).” We have also added this over-representation in our sample to the Strengths & Limitations section. “Purposive recruitment strategies were used to try and increase the variability of experiences within the group of participants interviewed, but despite this, our sample contained more 15- and 16-year-olds than other age groups, and participants who identified as White than other ethnic groups.”
Also, did you consider sexual orientation and gender identity? If not, please discuss is as a limitation.	Thank you for making this important point, which we have now acknowledged as a limitation of the research: “Those whom we spoke with were from a range of living situations and health statuses; however, there were some data we did not collect that is likely to have influenced participant experiences during the pandemic, including socioeconomic status, sexual orientation, and gender expression.”
You mention that “Those who returned a signed informed consent form also completed a demographics questionnaire” – not	Thank you for pointing out this unclear wording, which we have now clarified.

everyone returned a signed informed consent? If so, how did they consent to participate?	“All participants completed a demographics questionnaire (See Table 1 for participant demographics).” All participants provided consent (aged 16 and over) and assent with parental consent (aged under 15).
“All interested individuals were invited to contact the study team for further information” – how many people contacted you? How many did you select? How many actually agreed to be interviewed? Did you interview all who agreed?	We did not record the response rates during data collection, and are therefore, unable to provide data on several of these queries. This has been acknowledged in the Methods section on page 4 under Recruitment. We have now added some further text regarding the latter query that we are able to answer. “All those who were offered an interview agreed to take part.”
“Potential participants received a Study Information Sheet” – why and what was the content?	We have added further explanation to the wording of this sentence to explain the purpose of sending people the Participant Information Sheet: “In order to make an informed decision about whether to participate, potential participants received a Study Information Sheet with further details on the study, data protection measures and links to organisations providing mental health support. All were given the opportunity to ask questions.”
“Research interviews were the only source of data collected during the study” – this is a strange sentence. Why not saying “we used semi-structured interviews for data collection”?	We included this sentence because we used the SRQR reporting checklist featured on the EQUATOR network for enhancing the quality/transparency of qualitative research reporting (https://www.equator-network.org/reporting-guidelines/srqr/). Item 6 under the Methods section recommends that authors acknowledge sources of data in the study.
“We conducted data collection as two separate groups, to ensure any newly generated themes discussed by participants in each age group (as identified by the researcher) were sufficiently explored during data collection and analysis” – which were the two different groups? And how separating the groups helped to ensure that new themes were sufficiently explored?	We have now supplied further information on these points, to a) clarify the groups (which were adolescents aged 13-17 and young adults aged 18-24), And b) to explain why the groups were separated to begin with. “In order to capture some of the nuanced differences between experiences of adolescents compared with young adults, we made several key choices during data collection and analysis. We conducted data collection as two separate groups according to participant age (group 1: adolescents aged 13-17 years, group 2: young adults aged 18-24). We made this decision in order to focus on experiences that were specific to different stages of development in adolescence and “emerging” adulthood (Arnett, 2000). Where possible in our results, efforts have been made to clarify any similarities and differences in participant responses based on their age group. Rather than progressing in a linear fashion from stage to stage during qualitative data analysis, (Elliott, 2018) we made a decision to begin coding before data

	collection had finished. This approach allowed us to reflect on the way questions were asked during interviews and use prompts to learn more about topics of relevance to the research aims.”
Results	
Table 1: Do you have an explanation for why participants concentrated between 15 and 16yo?	Thank you for drawing this to our attention – We have added further text to specify that the sample was restricted based on who was willing to come forward. “Although we screened participants to ensure people from multiple groups were included, the nature of our recruitment process meant our sample was ultimately recruited by convenience (Campbell et al., 2020).” And “Purposive recruitment strategies were used to try and increase the variability of experiences within the group of participants interviewed, but despite this, our sample contained more 15- and 16-year-olds than other age groups, and participants who identified as White than other ethnic groups.”
In the quotes, why not informing participants exact age? It is quite different to be confined when you are 18 or 22.	Due to UK Data Protection Regulations, we are unable to provide the exact age of participants alongside their quotes to protect their identity. These are classified as “indirect identifiers” and therefore, including an age range of 5 years was a condition of publication as set out by the preprint server, MedRxiv: https://www.medrxiv.org/
Also, considering loneliness was a major struggle, did participants report how this affected their sex life and relationship with sexual partners? You briefly mention it in your results, but I was wondering if your data could allow you to develop it a bit further.	Although interviewers used an open-ended, non-judgemental and empathic method of questioning and made a concerted effort in developing rapport with participants, we did not learn enough about participants’ sex lives to comment on this in the findings. Some were using dating apps and others were in “long distance” relationships, as we have discussed. We have also added several more sentences on this to make it clearer. But otherwise, no other specifics were provided by participants on this topic. “Older participants in particular said that they were more focused on online dating during lockdown as a way to connect: “I joined a dating app just to meet people.” And others sought out dating apps as a way of alleviating boredom.” “I personally don’t think I would’ve downloaded that [dating] app if it hadn’t have been for the pandemic

	because I've tried them before and really didn't like them and hadn't had much success. And then I was just bored and felt like when am I next going to meet someone?... I feel like that probably wouldn't have happened if it wasn't for lockdown." P77, female, aged 20-24" We have also mentioned this topic in the Discussion. "Participants in our study frequently cited mental health consequences due to social isolation and loneliness, which some used gaming platforms, social media, and online dating apps to help address."
Generally speaking, your data show that the pandemic had paradoxical effects on mental health and well-being – while for some it led to feeling depressed and lonely, for other it increased mental health awareness and connexion to others. I wonder if you explored if there were any demographics associated with these different perceptions and experiences. There may not be, but since you present the demographics of the group, I think it would be worth exploring it.	Thank you very much for this thoughtful suggestion. We did consider this as the analysis was being undertaken and subsequently written up. Interestingly, other than what has been described, there were no obvious demographic or contextual characteristics associated with the cases who described increases in connection / MH awareness or loneliness. We have gone through the data again to ensure nothing was missed and still found this to be the case.
Discussion	
Both sections are clearly presented and reflecting the study results.	Many thanks for this feedback.
I suggest that you consider the limitations of not having included or identified certain groups (e.g., gay and transgender youth, those in poorer areas where schools did not manage to put online teaching in place, or who did not have the means to access this type of teaching) among which the literature shows that lockdown can have a more negative impact.	Thank you for this suggestion, which we have addressed with the following addition: "Those whom we spoke with were from a range of living situations and health statuses; however, there were some data we did not collect that is likely to have influenced participant experiences during the pandemic, including socioeconomic status, sexual orientation, and gender expression."
Also, maybe mentioning that other potentially important topics were not explored (e.g. sexual life), if that was the case.	We have now added the following text in response to this feedback: "We explored a range of issues during interviews that were pertinent to the research questions, but some topics identified during this study could be investigated in further depth by future researchers, including the impact of the COVID-19 pandemic on intimate relationships among adolescents and young adults."

VERSION 2 – REVIEW

REVIEWER	Scott, Stephanie Newcastle University
REVIEW RETURNED	12-Nov-2021

GENERAL COMMENTS	Overall, I am satisfied that the authors have responded to all queries raised - this is a well crafted paper and I'm looking forward to seeing it in print. I do have one point - which doesn't relate to my own original review points, but to including the age of participants in anonymised quotes. I'm inclined to agree with the reviewer - that there is a big difference between 13 and 17 year olds. I don't think age and gender with a quotation poses a risk to anonymity. My reading of GDPR and Data Protection is that an indirect identifier would need to be linked with another piece of personal data (place of work, job title, salary, their postcode or even the fact that they have a particular diagnosis or condition) to be problematic. This is of course a decision for the study team - but I have never encountered this rationale before for not including age with anonymised quotes.
---

REVIEWER	Ferraz, Dulce Fundação Oswaldo Cruz
REVIEW RETURNED	08-Dec-2021

GENERAL COMMENTS	Thanks for addressing all my comments. I look forward to seeing your paper published.
---